# Increase in Epithelial Permeability and Cell Metabolism by High Mobility Group Box 1, Inflammatory Cytokines and TPEN in Caco-2 Cells as a Novel Model of Inflammatory Bowel Disease

**DOI:** 10.3390/ijms21228434

**Published:** 2020-11-10

**Authors:** Maki Miyakawa, Takumi Konno, Takayuki Kohno, Shin Kikuchi, Hiroki Tanaka, Takashi Kojima

**Affiliations:** 1Sapporo IBD Clinic, Sapporo 064-0919, Japan; a030086m1982@yahoo.co.jp (M.M.); hirokit@mtc.biglobe.ne.jp (H.T.); 2Department of Cell Science, Research Institute for Frontier Medicine, Sapporo Medical University School of Medicine, Sapporo 060-8556, Japan; t.konno1225@gmail.com (T.K.); kohno@sapmed.ac.jp (T.K.); 3Department of Anatomy, Sapporo Medical University School of Medicine, Sapporo 060-8556, Japan; ksin@sapmed.ac.jp

**Keywords:** 2.5D culture, HMGB1, cytokines, TPEN, EW-7197, AG-1478, TNFα antibody, tight junctions, intestinal epithelial permeability, cell metabolism, cilia formation

## Abstract

High mobility group box 1 protein (HMGB1) is involved in the pathogenesis of inflammatory bowel disease (IBD). Patients with IBD develop zinc deficiency. However, the detailed roles of HMGB1 and zinc deficiency in the intestinal epithelial barrier and cellular metabolism of IBD remain unknown. In the present study, Caco-2 cells in 2D culture and 2.5D Matrigel culture were pretreated with transforming growth factor-β (TGF-β) type 1 receptor kinase inhibitor EW-7197, epidermal growth factor receptor (EGFR) kinase inhibitor AG-1478 and a TNFα antibody before treatment with HMGB1 and inflammatory cytokines (TNFα and IFNγ). EW-7197, AG-1478 and the TNFα antibody prevented hyperpermeability induced by HMGB1 and inflammatory cytokines in 2.5D culture. HMGB1 affected cilia formation in 2.5D culture. EW-7197, AG-1478 and the TNFα antibody prevented the increase in cell metabolism induced by HMGB1 and inflammatory cytokines in 2D culture. Furthermore, ZnSO_4_ prevented the hyperpermeability induced by zinc chelator TPEN in 2.5D culture. ZnSO_4_ and TPEN induced cellular metabolism in 2D culture. The disruption of the epithelial barrier induced by HMGB1 and inflammatory cytokines contributed to TGF-β/EGF signaling in Caco-2 cells. The TNFα antibody and ZnSO_4_ as well as EW-7197 and AG-1478 may have potential for use in therapy for IBD.

## 1. Introduction

High mobility group box 1 (HMGB1) is a chromatin-associated protein and one of the damage-associated molecular patterns (DAMPs) [1]. HMGB1 is abundantly and widely expressed in a variety of cell nuclei and plays a role in gene transcription in various human diseases [2]. HMGB1 promotes the induction of inflammatory cytokines in the pathogenesis of various inflammatory diseases [3]. Many recent studies have indicated that HMGB1 protein is involved in the development of inflammatory bowel disease (IBD) [4].

IBD is a chronic inflammatory condition of the gastrointestinal tract that presents not only digestive symptoms but also various general symptoms. It is immune-mediated and involves a complex interplay of genetic, immunological and environmental factors. The role of HMGB1 in the pathogenesis of IBD has been explored [4]. In an ulcerative colitis (UC) mouse model prepared with dextran sulfate sodium (DSS), colonic expression of HMGB1 and its receptor, RAGE, were significantly higher than in control mice [5]. Coincubation of the cytokines TNFα, IFNγ and IL-1β with human colon cancer Caco-2 cells and rat intestinal epithelial cells (IECs) revealed supernatant concentrations of HMGB1 [6]. Caco-2 enterocytic monolayer cells with HMGB1 or B-box increases NO expression and hyperpermeability of Caco-2 cells [7]. The amount of fecal HMGB1 correlates with mucosal inflammation and healing, including in IBD patients [8]. Thus, HMGB1 has the potential to be a therapeutic target for IBD [4], however, until now there have been no detailed reports on the effects of HMGB1 on the intestinal epithelial barrier function in IBD.

EW-7197 is a transforming growth factor-β type 1 receptor kinase inhibitor that has potentially useful therapeutic properties against colitis, with clinical translational potential for inhibiting key pathological responses of inflammation and fibrosis in patients with colitis [9]. AG-1478 is an epidermal growth factor receptor (EGFR) kinase inhibitor that, like the MAP kinase inhibitor U0126, prevents the disassembly of tight junction (TJ) transmembrane proteins caused by inflammatory cytokines such as, TNFα, IL-4 and IFNγ in Calu-3 airway epithelial cells [10]. We have previously reported that HMGB1-downregulated angulin-1/LSR induces epithelial barrier disruption via the TJ molecule claudin-2 (CLDN-2) in Calu-3 cells [11]. We have also reported that HMGB1 enhances epithelial permeability via p63/TGF-β signaling in lung and terminal bronchial epithelial cells [12]. However, it is not yet known whether EW-7197 or AG-1478 prevents the epithelial barrier disruption triggered by HMGB1 in intestinal epithelial cells. 

The TJ is an epithelial cell–cell junction that regulates the flow of solutes through paracellular pathways and maintains cell polarity [13]. TJs are involved with various signal transduction pathways that regulate epithelial cell proliferation, gene expression, differentiation and morphogenesis [14]. Claudins (CLDNs) are major components of TJs in epithelial and endothelial cells [15]. Tricellular TJs (tTJs), comprised of angulin-1/ lipolysis-stimulated lipoprotein receptor (LSR) and tricellulin (TRIC), form at the convergence of bicellular TJs (bTJs) [16]. The tTJ molecule angulin-1/LSR contribute to the epithelial barrier [17,18]. Many studies have indicated that TJ molecules play important roles in the intestinal epithelial barrier of IBD [19]. In addition, an abundance of CLDN-2 has been widely observed in UC and Crohn’s disease (CD) [20,21]. TNFα and IL-13 have been linked to the observed claudin-2 increase [20,21]. TNFα also causes a dislocalization of CLDN-5 and CLDN-8 from the TJ to sub-TJ membrane components and into endosomes [21]. CLDN-4 and -7 are downregulated in UC [22]. In CD, CLDN-5 and -8 are downregulated [20]. In IBD, recent studies have indicated that zinc is important for the maintenance of the mucosal barrier [23]. A lack of zinc leads to the downregulation of occludin (OCLN) and zonula occludens-1 (ZO-1) proteins, which are TJ proteins, resulting in decreased TJ in Caco-2 cells [24]. Zinc supplementation increases transepithelial electrical resistance (TEER) and induces expression of CLDN-2, -7 and ZO-1 proteins [25,26]. However, the detailed mechanisms by which HMGB1 and zinc affect TJ molecules via tTJ in the intestinal epithelial barrier of IBD have not been elucidated.

In this study, we used Caco-2 cells to investigate how HMGB1 affects the dysfunction of intestinal epithelial barrier function. We analyzed Caco-2 monolayer cells (2D culture) and Caco-2 spheroid cells (2.5D Matrigel culture), which have lumen formation, with the aim of developing a new therapy for intestinal inflammation and injury in patients with IBD.

## 2. Results

### 2.1. HMGB1 Impairs Epithelial Barrier Function, and EW-7197, AG-1478 and TNFα-Antibody Prevent the Impairment by HMGB-1 in 2.5D Matrigel Culture of Caco-2 Cells

Our group previously reported that HMGB1 disrupts the epithelial barrier via downregulation of LSR in airway Calu-3 cells [11]. To investigate the effects of HMGB1 on Caco-2 cells, we treated the Caco-2 spheroid cells with 100 ng/mL HMGB1 for 24 h [12]. Spheroid cells were treated with 10 μM EW-7197, 10 μM AG-1478 or 40 μg/mL TNFα ab before treatment with 100 ng/mL HMGB1. Treatment with HMGB1 induced the permeability of FD-4 into the lumina of 8 of 10 spheroids, whereas treatment with EW-7197, AG-1478 or the TNFα-antibody prevented the hyperpermeability of FD-4 induced by HMGB1 into the lumina of 9 of 10 spheroids (Figure 1A). We then measured FD-4 intensity for quantification. The value was increased by the treatment with HMGB1, whereas treatment with EW-7197, AG-1478 or the TNFα-antibody prevented the increase in values induced by HMGB1 (Figure 1B). Immunocytochemical analysis showed that the treatment with HMGB1 decreased LSR at the membranes, while OCLN was detected at the luminal surfaces of 7 of 10 spheroids. Treatment with EW-7197, AG-1478 or the TNFα ab prevented the changes in expression of TJs caused by HMGB1 in 8 of 10 spheroids (Figure 2A). Transmission electron microscopic (TEM) analysis revealed that “kissing points”, indistinguishable distances between two adjacent cell membranes, were loosened by treatment with HMGB1 in spheroid cells, and this change was prevented by treatment with EW-7197 (Figure 2B). Western blotting of the 2.5D Matrigel culture showed that HMGB1 decreased the expression of TRIC and CLDN-1, and treatment with EW-7197 or AG-1478 prevented the change in expression induced by HMGB1 (Figure 2C).

### 2.2. TNFα and IFNγ Impair the Epithelial Barrier Function, and EW-7197, AG-1478 and TNFα-Antibody Prevent the Impairment by TNFα and IFNγ in 2.5D Matrigel Culture of Caco-2 Cells

To investigate the effects of TNFα and IFNγ on the 2.5D Matrigel culture of Caco-2 cells, we treated the Caco-2 spheroid cells with 100 μg/mL TNFα and 100 µg/mL IFNγ for 24 h [11]. Some spheroid cells were treated with 10 μM EW-7197, 10 μM AG-1478 or 40 μg/mL TNFα ab before treatment with 100 μg/mL TNFα and 100 μg/mL IFNγ. Treatment with TNFα and IFNγ induced the permeability of FD-4 into the lumina of 8 of 10 spheroids, whereas treatment with EW-7197, AG-1478 or the TNFα-antibody prevented the hyperpermeability of FD-4 into the lumina of 7 of 10 spheroids induced by TNFα and IFNγ (Figure 3A). We measured the FD-4 intensity for quantification. The value was increased by the treatment with TNFα and IFNγ, whereas treatment with EW-7197, AG-1478 or the TNFα-antibody prevented the increase in values caused by TNFα and IFNγ (Figure 3B). Immunocytochemistry revealed that the treatment with TNFα and IFNγ decreased LSR at the membranes, while OCLN was detected at the luminal surfaces of 8 of 10 spheroids. Treatment with EW-7197, AG-1478 or the TNFα-ab prevented the changes in expression of TJs caused by TNFα and IFNγ in 7 of 10 spheroids (Figure 4A). The same results were obtained by treatment with IL-1β or IL-13 (Appendix A). Western blotting of the 2.5D Matrigel culture showed that TNFα and IFNγ decreased the expression of TRIC and CLDN-1, and the treatment with EW-7197 or AG-1478 prevented the change in expression induced by treatment with TNFα and IFNγ (Figure 4B).

### 2.3. HMGB1 Affected Cilia Formation and EW-7197 or AG-1478 Prevented its Effects in 2.5D Matrigel Culture of Caco-2 Cells

To investigate the effects of HMGB1 on cilia formation, we performed immunocytochemistry and TEM analyses. Immunocytochemistry revealed that the treatment with HMGB1 increased PAR3 and decreased Ac-tub at the luminal surfaces of 8 of 10 spheroids. Treatment with EW-7197 or AG-1478 prevented these changes caused by HMGB1 in 8 of 10 spheroids (Figure 5A). The same results were obtained by the treatment with TNFα and IFNγ (Appendix A). TEM analysis showed that the number of cilia was decreased by treatment with HMGB1 and this change was prevented by treatment with EW-7197 (Figure 5B). 

### 2.4. TPEN Impairs Epithelial Barrier Function, and ZnSO_4_ Prevents the Impairment by HMGB-1 in 2.5D Matrigel Culture of Caco-2 Cells

Zinc deficiency is associated with an increased risk of subsequent hospitalization, surgery, and disease-related complications in patients with IBD, Crohn’s disease (CD), and UC [27]. To investigate how zinc deficiency affected the 2.5D Matrigel culture of Caco-2 cells, we treated the Caco-2 spheroid cells with 200 nM ZnSO_4_ or 10 nM TPEN for 24 h [26]. Some spheroid cells were treated with 200 nM ZnSO_4_ before treatment with 10 nM TPEN. Treatment with TPEN induced the permeability of FD-4 into the lumina of 7 of 10 spheroids, whereas treatment with ZnSO_4_ prevented the hyperpermeability of FD-4 induced by TPEN in the lumina of 8 of 10 spheroids (Figure 6A). We measured FD-4 intensity for quantification. The value was increased by the treatment with TPEN, whereas treatment with ZnSO_4_ prevented the increase in values induced by TPEN (Figure 6B). Immunocytochemistry showed that the treatment with TPEN decreased LSR and Ac-tub at the membranes, while OCLN was detected at the luminal surfaces of 7 of 10 spheroids. Treatment with ZnSO_4_ prevented the changes in expression of TJs caused by TPEN in 8 of 10 spheroids (Figure 7A). Western blotting of the 2.5D Matrigel culture showed that TPEN decreased the expression of LSR, TRIC CLDN-1 and pMAPK, whereas it increased the expression of CLDN-4. The treatment with ZnSO_4_ increased the expression of LSR and pMAPK, and prevented the change in expression caused by treatment with TPEN (Figure 7B).

### 2.5. Low Glucose Condition Impairs Epithelial Barrier Function in 2.5D Matrigel Culture of Caco-2 Cells

It has been reported that IBD increases the risk of type 2 diabetes [28]. Therefore, it is considered that the glucose condition may affect intestinal barrier function. To investigate how the low glucose condition affects the 2.5D Matrigel culture of Caco-2 cells, we treated the Caco-2 spheroid cells with low glucose DMEM medium for 24 h. Treatment with low glucose DMEM medium induced the permeability of FD-4 into the lumina of 8 of 10 spheroids (Figure 8A). We measured FD-4 intensity for quantification. The value was increased by the treatment with low glucose DMEM medium (Figure 8B). By immunocytochemistry, the treatment with low glucose DMEM medium was shown to decrease LSR and Ac-tub at the membranes, whereas OCLN was detected at the luminal surfaces of 8 of 10 spheroids (Figure 8C). Western blotting of the 2.5D Matrigel culture showed that low glucose DMEM medium decreased the expression of LSR, TRIC and pMAPK, whereas it increased the expression of pAMPK (Figure 8D).

### 2.6. Low Glucose Condition Decreases Mitochondrial Respiration Levels in 2D Culture of Caco-2 Cells

To investigate whether the glucose condition affected the cell metabolism in Caco-2 cells, we treated Caco-2 cells in 2D culture with low glucose DMEM for 24 h and measured OCR. This treatment decreased baseline OCR, proton leak and ATP production compared to control medium (high glucose DMEM) (Figure 9A,B).

### 2.7. HMGB1 Enhances Mitochondrial Respiration Levels in 2D Culture of Caco-2 Cells

To investigate whether cell metabolism was altered by HMGB1 and whether EW-7197 or AG-1478 could prevent the change, Caco-2 cells in 2D culture were pretreated with 10 μM EW-7197 or 10 μM AG-1478 before treatment with 100 ng/mL HMGB1 for 24 h and measured for OCR. Treatment with HMGB1 increased baseline OCR, proton leak and ATP production. EW-7197 and AG-1478 prevented these changes caused by treatment with HMGB1. TNFα-ab also prevented these changes (Figure 9B).

### 2.8. TNFα and IFNγ Enhance Mitochondrial Respiration Levels in 2D Culture of Caco-2 Cells

To investigate whether cell metabolism was altered by TNFα and IFNγ and whether EW-7197, AG-1478 or the TNFα-ab could prevent the change, Caco-2 cells in 2D culture were pretreated with 10 μM EW-7197, 10 μM AG-1478 or 40 μg/mL TNFα ab before treatment with 100 μg/mL TNFα and 100 μg/mL IFNγ for 24 h and measured for OCR. Treatment with TNFα and IFNγ increased baseline OCR and ATP production. EW-7197 and AG-1478 prevented these changes induced by treatment with TNFα and IFNγ. TNFα-ab did not prevent these changes (Figure 9B).

### 2.9. TPEN and ZnSO_4_ Affects Mitochondrial Respiration Levels in 2D Culture of Caco-2 Cells

To investigate whether cell metabolism was altered by TPEN and ZnSO_4_, Caco-2 cells in 2D culture were treated with 200 nM ZnSO_4_ or 10nM TPEN for 24 h and measured for OCR. Some Caco-2 cells were pretreated with ZnSO_4_ before treatment with TPEN. Treatment with ZnSO_4_ or TPEN also increased baseline OCR and ATP production. However, TPEN prevented the change induced by treatment with ZnSO_4_ (Figure 9B). 

### 2.10. Expression Patterns of LSR and HMGB1 in the Colonic Epithelium of Inflammatory Bowel Disease

We performed immunohistochemical analysis for LSR and HMGB1 in the normal colonic epithelium and that of IBD. The immunohistochemical results for IBD showed that in the ductal structural area of regenerative colonic epithelium, the expression of HMGB1 was higher than that of the normal region. LSR was not detected in regenerative colonic epithelium (Figure 10, Appendix A).

## 3. Discussion

In the present study, HMGB1 and inflammatory cytokines induced epithelial permeability and cell metabolism with downregulation of the tight junction molecules LSR, TRIC and CLDN-1 in 2.5D culture of Caco-2 cells.

HMGB1 is a proinflammatory mediator belonging to the alarmin family [2]. HMGB1-dowregulation of angulin-1/LSR induces epithelial barrier disruption via the upregulation of CLDN-2 expressed in the tight junctions of leaky epithelia in airway epithelial cell line Calu-3 [11]. Loss of LSR affects the epithelial barrier integrity in Caco-2 cells [29]. The fecal level of HMGB1 is correlated with mucosal inflammation [8]. Infusing HMGB1 in the colon of mice resulted in a two- to three-fold increase in the release of LDH, which is a marker of epithelial damage, further indicating that intraluminal HMGB1 induces tissue damage in the colon [30]. In this study, TNFα + IFNγ also induced epithelial permeability of Caco-2 cells with downregulation of LSR, TRIC and CLDN-1 in 2.5D cultures. Moreover, IL-1β and IL-13 also induced epithelial permeability of Caco-2 cells in 2.5D cultures. It is well known that many proinflammatory cytokines, such as TNFα, IFNγ, IL-1β and IL-13, are involved in the pathogenesis of IBD and these proinflammatory cytokines affect the epithelial barrier and TJ molecules [19]. We used this 2.5D Caco-2 cell culture model as an IBD model for analyzing further effects of proinflammatory cytokines. Our findings suggest that HMGB1 disrupts the intestinal epithelial barrier and induces permeability via downregulation of the TJ molecules LSR, TRIC and CLDN-1 in a manner similar to other proinflammatory cytokines. These changes induced by HMGB1 lead to loosening of the tight junction strands of intestinal epithelial cells.

In this study, the TGF-β type 1 receptor kinase inhibitor EW-7197 and EGFR kinase inhibitor AG-1478 prevented the changes in the hyperpermeability and expression of TJ molecules induced by HMGB1, TNFα + IFNγ, IL-1β and IL-13. EW-7197 is a transforming growth factor-β type 1 receptor kinase inhibitor with potential anti-inflammatory and antifibrotic properties. It prevents ulcerative colitis-associated fibrosis and inflammation as well as changes in the distribution of angulin-1/LSR and the epithelial barrier function caused by TGF-β in a pancreatic cancer cell line [9,31]. AG-1478 is an epidermal growth factor receptor (EGFR) kinase inhibitor, and the disassembly of tight junction (TJ) transmembrane proteins caused by inflammatory cytokines, such as TNFα, IL-4 and IFNγ is prevented by it and the MAP kinase inhibitor U0126 in Calu-3 airway epithelial cells [10]. In Caco-2 cells, L-glutamine reduces the acetaldehyde-induced redistribution of occludin, ZO-1, E-cadherin, and beta-catenin from the intercellular junctions and induces a rapid increase in the tyrosine phosphorylation of EGF receptors. The protective effect of L-glutamine is prevented by AG-1478 [32]. Moreover, the TNFα-ab adalimumab also prevents the hyperpermeability induced by these proinflammatory cytokines. Adalimumab is a well-known anti-TNFα drug used for treatment of IBD patients [33]. In Caco-2 cells, adalimumab prevents increased phosphorylation of the myosin light chain and reverses the TNF-induced downregulation of CLDN-1 and -4 [34]. These findings suggest that EW-7197, AG-1478 and TNFα-ab prevent the hyperpermeability induced by HMGB1 and inflammatory cytokines in intestinal epithelial cells. EW-7197 and AG-1478 may have protective roles against the disruption of tight junctions by inflammatory cytokines.

HMGB1 affects not only the epithelial barrier function but also other cellular functions. In human nasal epithelial cells, the depletion of sirtuin 6 (Sirt6), one of the sirtuin family members that is widely studied in aging, DNA repair, metabolism, inflammation and cancer, suppresses the number of human nasal epithelial cell cilia and dramatically induces HMGB1 translocation from the nucleus to cytoplasm [35]. In this study, the expression pattern of the epithelial polarity protein PAR3, normally present at the apical-lateral membrane of basal cells, was dispersed and the length of cilia was shortened by HMGB1 in 2.5D cultures of Caco-2 cells. These changes induced by HMGB1 were prevented by EW-7197 and AG-1478. PAR3 is co-localized with LSR at the centrosome [36]. These findings suggest that HMGB1 may affect the intestinal epithelial cell polarity and cilia formation via the downregulation of LSR.

The absorption of zinc in patients with IBD is lower than in healthy controls [23,27,37]. In Caco-2 cells, zinc deficiency leads to the downregulation of OCLN and ZO-1, and zinc supplementation increases TEER and induces expression of CLDN-2, -7 and ZO-1 [24,25,26]. In this study, zinc deficiency caused by treatment with both TPEN and HMGB1 disrupted the epithelial barrier and induced permeability via downregulation of TJ molecules LSR, TRIC, CLDN-1 and pMAPK in Caco-2 cells in 2.5D cultures. These changes were prevented by zinc supplementation via ZnSO_4_ treatment. These findings suggest that zinc may play an important role in the regulation of intestinal epithelial barrier function via tight junction molecules and MAPK signaling.

In this study, the low glucose condition induced hyperpermeability via downregulation of LSR, TRIC, CLDN-1 and pMAPK with upregulation of pAMPK in 2.5D Caco-2 cell culture. Moreover, the low glucose condition decreased OCR in Caco-2 cells. These findings may suggest that the low glucose condition disrupts the epithelial barrier function via downregulation of TJ molecules and OCR in intestinal epithelial cells. However, it was recently reported that IBD increases the risk of type 2 diabetes [28]. It has been reported that high glucose treatment results in reduced cell viability, increased reactive oxygen species production, measurable mitochondrial dysfunction, and decreased transepithelial electrical resistance, with increased membrane permeability in Caco-2 cells [38]. Considering these findings, the high glucose condition may induce the secretion of proinflammatory cytokines and cause inflammatory conditions, leading to downregulation of epithelial barrier function. Thus, further analyses are necessary to elucidate the connection between the glucose condition and inflammatory condition. It was reported that HMGB1 increases OCR and decreases the epithelial barrier function in airway epithelial cell line Calu-3 [11]. In this study, treatments with inflammatory cytokines increased OCR and decreased the epithelial barrier function in Caco-2 cells, and these changes were prevented by EW-7197 and AG-1478. These findings indicate that there is a relationship between OCR affected by proinflammatory cytokines and the epithelial barrier function.

In conclusion, our findings indicate that EW-7197 and AG-1478 prevented hyperpermeability induced by HMGB1 and inflammatory cytokines in 2.5D culture of Caco-2 cells. We previously reported that EW-7197 also prevented hyperpermeability induced by HMGB1 in 2.5D culture of human lung epithelial cells [12]. Intestinal injury can lead to lung injury via HMGB1 released from the intestinal injury [39]. EW-7197 may be useful as a novel therapy for inflammatory diseases involving the lung and gut. Furthermore, Caco-2 cells in 2.5D Matrigel culture are similar to human intestinal organoids in vivo and may be a useful in vitro model for studying IBD and other intestinal diseases. 

## 4. Materials and Methods

### 4.1. Antibodies and Reagents

A rabbit polyclonal anti-tricellulin (TRIC) antibody was obtained from Zymed Laboratories (San Francisco, CA, USA). The rabbit polyclonal anti-lipolysis-stimulated lipoprotein receptor (LSR) antibody was from Novus Biologicals (Littleton, CO, USA). Rabbit polyclonal anti-claudin (CLDN)-1, CLDN-4, CLDN-7 and mouse monoclonal anti-occludin (OCLN) (Clone OC-3F10) and anti-acetylated tubulin (Ac-tub) antibodies were from Zymed Laboratories (San Francisco, CA, USA). Anti-phosphorylated MAPK (pMAPK), AMPK (pAMPK) and Smad2/3 (pSmad2/3) antibodies were from Cell Signaling Technology (Danvers, MA, USA). Recombinant human HMGB1, TNFα, IFNγ, Interleukin (IL)-1β and IL-13 were from Sigma-Aldrich (St. Louis, MO, USA). The TGF-β receptor type 1 inhibitor EW-7197 (N-(4-([1,2,4] triazolo [1,5-a] pyridin-6-yl)-5-(6-methylpyridin-2-yl)-1H-imidazol-2-yl) methyl)-2-fluoroaniline) was obtained from Cayman Chemical (Ann Arbor, MI, USA). The EGF receptor tyrosine kinase inhibitor AG-1478 (4-[3-chlorophenyl methyl]-6,7-dimethoxyquinazoline) was from Abcam (Cambridge, MA, USA). The anti-human-TNFα mAb adalimumab (TNFα ab) (ADA, Humira^®^) was from Eisai Co., Ltd. (Tokyo, Japan). TPEN (N,N,N′,N′-tetrakis [2-pyridylmethyl] ethylenediamine) was from Abcam (Cambridge, MA, USA). ZnSO_4_ was purchased from Sigma-Aldrich (St. Louis, MO, USA). The rabbit polyclonal anti-actin antibody was from Sigma-Aldrich (St. Louis, MO, USA). Alexa 488 (green)-conjugated anti-rabbit IgG and Alexa 594 (red)-conjugated anti-mouse IgG antibodies and Alexa 594 (red)-conjugated phalloidin were from Molecular Probes, Inc. (Eugene, OR, USA). HRP-conjugated polyclonal goat anti-rabbit IgG was from Dako A/S (Glostrup, Denmark). FITC-dextran (FD-4, MW 4.0 kDa) was from Sigma-Aldrich Co. (St. Louis, MO, USA). 

### 4.2. Cell Line Culture and Treatment 

The human colorectal adenocarcinoma cell line Caco-2 (ATCC^®^ HTB-37^™^) was purchased from American Type Culture Collection (ATCC) (Manassas, VA, USA). Caco-2 cells were maintained in high glucose Dulbecco’s Modified Eagle Medium (DMEM: Sigma-Aldrich; Merck Millipore) supplemented with 10% dialyzed fetal bovine serum (FBS; Thermo Fisher Scientific, Waltham, MA, USA). The medium contained 100 U/mL penicillin, 100 μg/mL streptomycin, 50 μg/mL amphotericin-B and 4.5 g/L glucose. The cells were plated on 35 and 60-mm culture dishes coated with rat tail collagen (500 μg dried tendon/mL in 0.1% acetic acid) and incubated in a humidified 5% CO_2_ incubator at 37 °C. Some cells were treated with low glucose Dulbecco’s Modified Eagle Medium (DMEM: Sigma-Aldrich; Merck Millipore) supplemented with 10% dialyzed fetal bovine serum (FBS; Thermo Fisher Scientific, Waltham, MA, USA). This medium contained 100 U/mL penicillin, 100 μg/mL streptomycin, 50 μg/mL amphotericin-B and 1.0 g/L glucose. Other cells were treated with 100 ng/mL HMGB1, 10 μM EW-7197, 10 μM AG-1478, 40 µg/mL TNFα ab, 100 µg/mL TNFα, 100 µg/mL IFNγ, 200 nM ZnSO_4_ and 10 nM TPEN.

### 4.3. 2.5-Dimensional Matrigel Culture 

Thirty-five millimeter culture dishes or 35-mm culture glass-coated dishes were coated with 100% Matrigel (30 μL or 15 μL) at 4 °C and incubated at 37 °C for 30 min. Caco-2 cells (5 × 10^4^) were plated in high glucose Dulbecco’s Modified Eagle Medium (DMEM: Sigma-Aldrich; Merck Millipore) supplemented with 10% dialyzed fetal bovine serum (FBS; Thermo Fisher Scientific, Inc., Waltham, MA, USA). After 24 h of plating, the cells were pretreated with 10 μM EW-7197. HMGB1. In all experiments, ten spheroids were examined.

### 4.4. Immunocytochemistry 

Caco-2 spheroids were fixed with cold acetone and ethanol (1:1) at −20 °C for 10 min. After rinsing in PBS, the cells were incubated with anti-OCLN, anti-LSR, anti-TRIC, anti-PAR3 and anti-Ac-tub antibodies (1:100) overnight at 4 °C. Alexa Fluor 488 (green)–conjugated anti-rabbit IgG and Alexa Fluor 592 (red)–conjugated anti-mouse IgG were used as secondary antibodies. Fluorescence images were acquired by an Olympus IX 71 inverted microscope (Olympus Co.; Tokyo, Japan) and a confocal laser scanning microscope (LSM5 PASCAL; Carl Zeiss, Jena, Germany). 

### 4.5. Fluorescein Isothiocyanate (Fitc) Permeability Assay 

To assess barrier function, the permeability of fluorescein isothiocyanate (FITC)-dextran (FD-4, MW 4.0 kDa) from the outside into the spheroid lumen was examined by using 2.5D Matrigel culture of HLE cells on 35-mm glass-coated dishes. These cultured Caco-2 cells were incubated in the medium with 1% FD-4 at 37 °C for 2 h. In all experiments, 10 spheroids were photographed and measured using confocal laser scanning microscope with imaging software (LSM5 PASCAL; Carl Zeiss, Jena, Germany). 

### 4.6. Transmission Electron Microscopic Analysis 

Caco-2 spheroids were cultured to confluence in 8 chambers of CultureSlides (FALCON). For TEM, the cultured cells were fixed in 2.5% glutaraldehyde in PBS overnight at 4 °C, followed by post-fixing in 2% osmium tetroxide in the same buffer. The ultrathin sections were stained with uranyl acetate followed by lead citrate and examined at 80 kV with a transmission electron microscope (H7500; Hitachi, Tokyo, Japan).

### 4.7. Western Blot Analysis 

Caco-2 spheroids were scraped from a 35 mm dish containing 400 μL of buffer (1 mM NaHCO3 and 2 mM phenylmethylsulfonyl fluoride). The samples were separated by electrophoresis in 520% SDS polyacrylamide gels (Wako, Osaka, Japan), and electrophoretically transferred to a nitrocellulose membrane (Immobilon; Millipore Co.; Bedford, UK). The membrane was incubated with anti-LSR, anti-TRIC, anti-CLDN-1, anti-CLDN-4, anti-CLDN-7, anti-pSmad2/3, anti-pMAPK, anti-pAMPK and anti-actin antibodies (1:1000) at 4 °C overnight. Then it was incubated with an HRP-conjugated anti-rabbit IgG antibody at room temperature for 1 h. The immunoreactive bands were detected using an ECL Western blotting system (GE Healthcare, Little Chalfont, UK). They were quantitated by densitometry and the data normalized to actin. 

### 4.8. Immunohistochemical Analysis 

Human colon tissues were obtained from 8 patients with Crohn’s disease (CD) and 8 with ulcerative colitis (UC) who underwent colectomy at Sapporo Kosei General Hospital. This study was approved by the Ethics Committee of the hospital. The tissues were embedded in paraffin after fixation with 10% formalin in PBS. The tissue sections were then washed twice with Tris-buffered saline (TBS) and pre-blocked with Block Ace for 1 h. After washing with TBS, the sections were incubated with anti-LSR (1:100) and anti-HMGB1 (1:400) antibodies for 1 h. They were incubated using Vision BioSystems Bond Polymer Refine Detection kit DS9800. The sections were counterstained with hematoxylin. 

### 4.9. XF96 Extracellular Flux Measurements

Mitochondrial respiration was assessed using an XF96 Extracellular Flux Analyzer (Aligent, Santa Clara, CA, USA). Caco-2 cells were seeded on XF96 plates at a density of 20,000 cells/well. One day prior to the experiment, sensor cartridges were hydrated with XF calibrate solution (pH 7.4) and incubated at 37 °C in a non-CO_2_ incubator for 24 h. Baseline measurements of mitochondrial respiration (OCR) were taken before sequential injection of the following inhibitors: 1 μM oligomycin A, 2 μM FCCP, and 1 μM antimycin A and rotenone.

### 4.10. Data Analysis 

Each set of results shown is representative of at least three separate experiments. Results are given as means ± SEM. Statistical analysis was conducted by one-way analysis of variance (ANOVA) with the Tukey-Kramer method. Statistical significance was set at ** *p* < 0.01 and ## *p* < 0.01.

### 4.11. Ethics statement 

The protocol for human study was reviewed and approved by the ethics committees of the Sapporo Medical University School of Medicine and Sapporo Kosei General Hospital. All experiments were carried out in accordance with the approved guidelines and the Declaration of Helsinki.

## Figures and Tables

**Figure 1 ijms-21-08434-f001:**
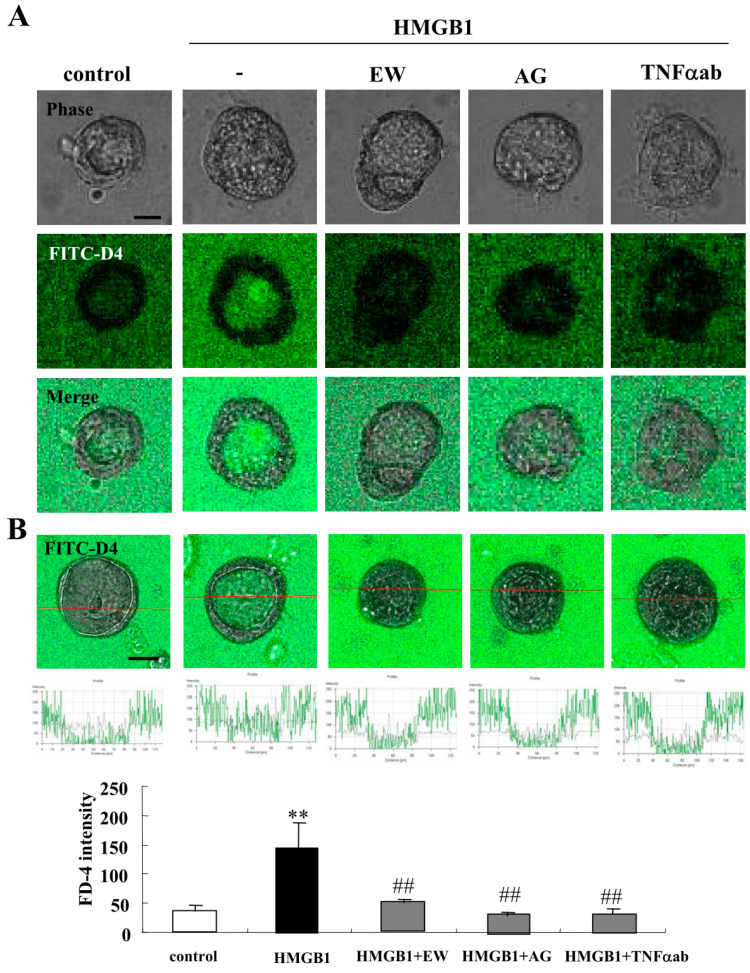
Effects of high mobility group box 1 protein (HMGB1) treatment on epithelial permeability in 2.5D Matrigel culture of Caco-2 cells. (**A**) Phase-contrast images and FD-4 assay of 2.5D Matrigel culture of Caco-2 cells pretreated with 10 μM EW-7197, 10 μM AG-1478 or 40 μg/mL TNFα ab before treatment with 100 ng/mL HMGB1. Scale bar: 20 μm. (**B**) Quantification of FD-4 intensity. Bar graph FD-4 intensity values representing barrier function of Caco-2 spheroids pretreated with 10 μM EW-7197, 10 μM AG-1478 or 40 μg/mL TNFα ab before treatment with 100 ng/mL HMGB1. ** *p* < 0.01, vs. control, ## *p* < 0.01, vs. HMGB1. Scale bar: 20 μm.

**Figure 2 ijms-21-08434-f002:**
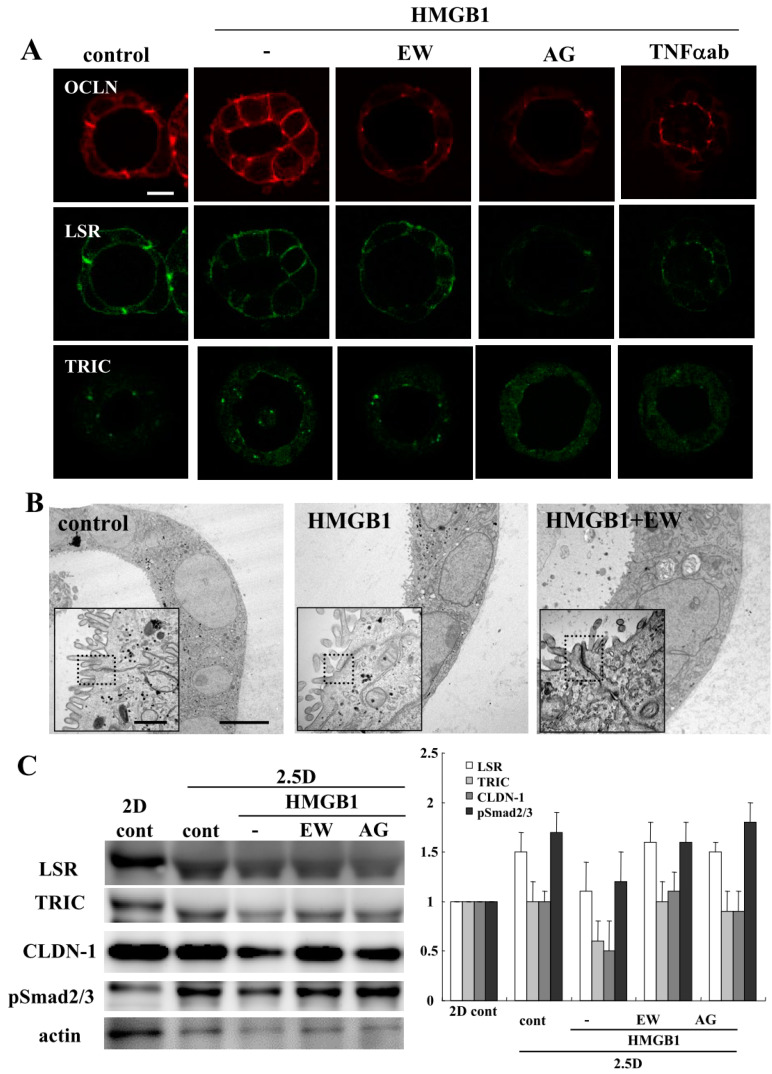
Effects of HMGB1 treatment on tight junction molecules in 2.5D Matrigel culture of Caco-2 cells. (**A**) Immunocytochemistry for occludin (OCLN), lipolysis-stimulated lipoprotein receptor (LSR) and tricellulin (TRIC) in 2.5D Matrigel culture of Caco-2 cells pretreated with 10 μM EW-7197, 10 μM AG-1478 or 40 μg/mL TNFα ab before treatment with 100 ng/mL HMGB1. Scale bar: 20 μm. (**B**) Transmission electron microscopic (TEM) analysis of Caco-2 spheroids treated with or without 10 μM EW-7197 before treatment with 100 ng/mL HMGB1. Scale bar: 2 μm. (**C**) Western blotting for LSR, TRIC, CLDN-1, pSmad 2/3 and actin in 2.5D Matrigel culture of Caco-2 cells pretreated with 10 μM EW-7197 or 10 μM AG-1478 before treatment with 100 ng/mL HMGB1. The corresponding expression levels of (**C**) are shown as a bar graph.

**Figure 3 ijms-21-08434-f003:**
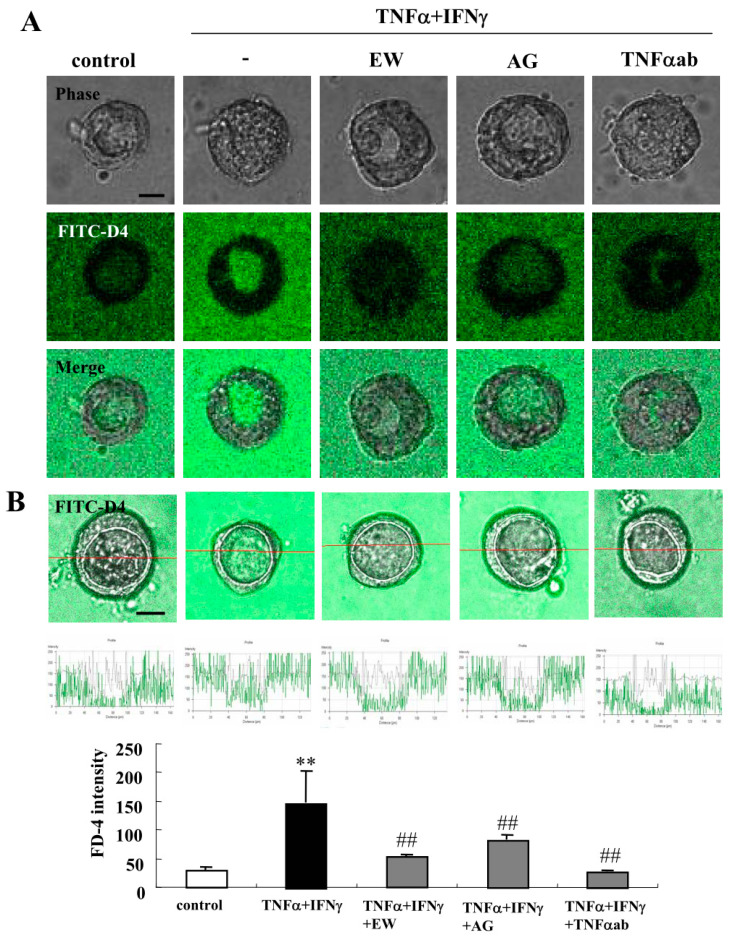
Effects of treatment with TNFα and IFNγ treatment on epithelial permeability in 2.5D Matrigel culture of Caco-2 cells. (**A**) Phase-contrast images and FD-4 assay of 2.5D Matrigel culture of Caco-2 cells pretreated with 10 μM EW-7197, 10 μM AG-1478 or 40 μg/mL TNFα ab before treatment with 100 μg/mL TNFα and 100 μg/mL IFNγ. Scale bar: 20 μm. (**B**) Quantification of FD-4 intensity. Bar graph FD-4 intensity values representing barrier function of Caco-2 spheroids pretreated with 10 μM EW-7197, 10 μM AG-1478 or 40 μg/mL TNFα ab before treatment with 100 μg/mL TNFα and 100 μg/mL TNFα. ** *p* < 0.01, vs. control, ## *p* < 0.01, vs. TNFα and IFNγ. Scale bar: 20 μm.

**Figure 4 ijms-21-08434-f004:**
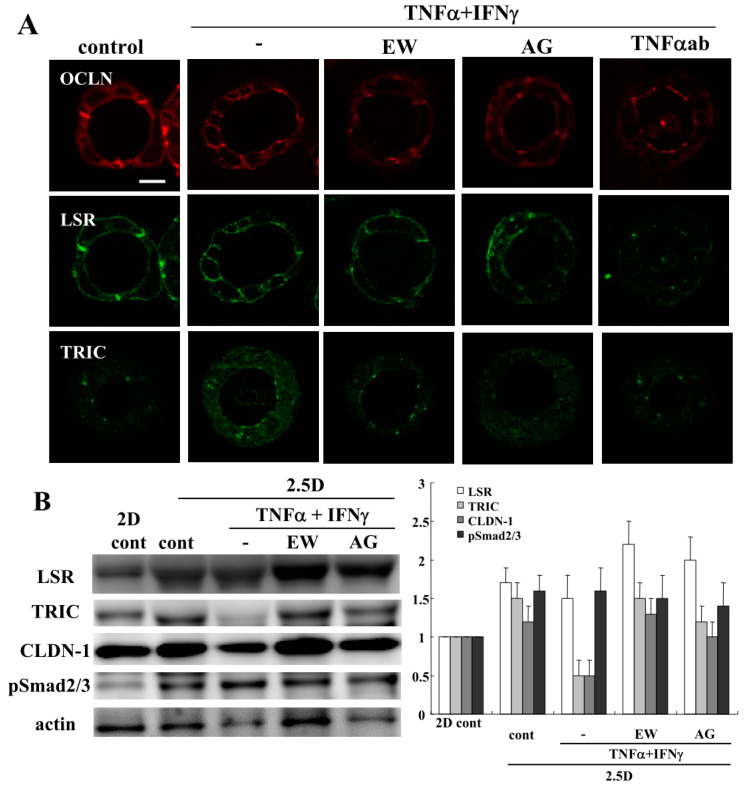
Effects of treatment with TNFα and IFNγ on tight junction molecules in 2.5D Matrigel culture of Caco-2 cells. (**A**) Immunocytochemistry for OCLN, LSR and TRIC in 2.5D Matrigel culture of Caco-2 cells pretreated with 10 μM EW-7197, 10 μM AG-1478 or 40 μg/mL TNFα ab before treatment with 100 μg/mL TNFα and 100 μg/mL IFNγ. Scale bar: 20 μm. (**B**) Western blotting for LSR, TRIC, CLDN-1, pSmad 2/3 and actin in 2.5D Matrigel culture of Caco-2 cells pretreated with 10 μM EW-7197 or 10 μM AG-1478 before treatment with 100 μg/mL TNFα and 100 μg/mL IFNγ. The corresponding expression levels of (**B**) are shown as a bar graph.

**Figure 5 ijms-21-08434-f005:**
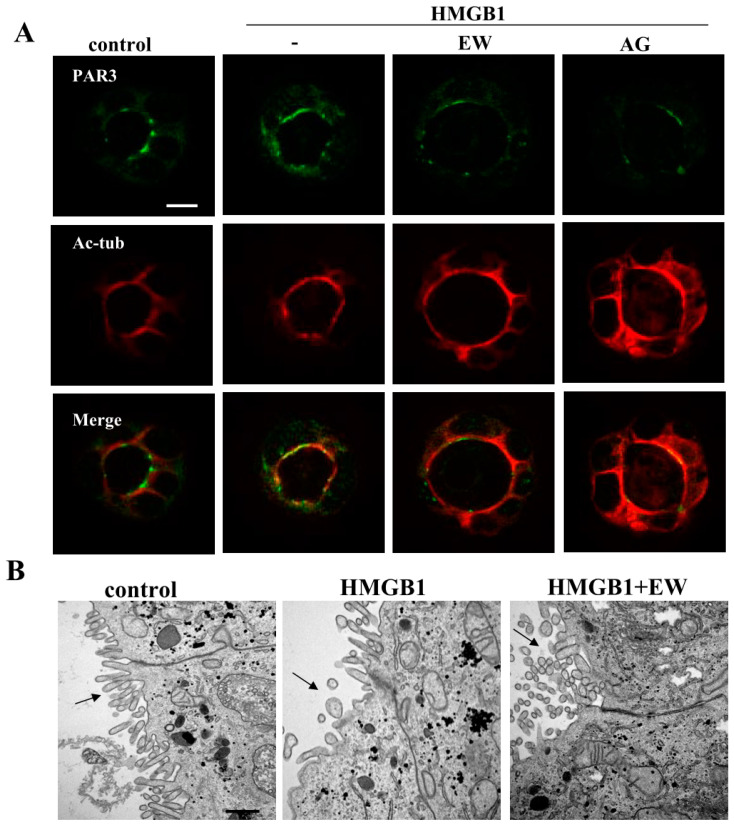
Effects of treatment with HMGB1 on cilia formation in 2.5D Matrigel culture of Caco-2 cells. (**A**) Immunocytochemistry for PAR3 and Ac-tub in 2.5D Matrigel culture of Caco-2 cells pretreated with 10 μM EW-7197 or 10 μM AG-1478 before treatment with 100 ng/mL HMGB1. Scale bar: 20 μm. (**B**) TEM analysis of Caco-2 spheroids treated with or without 10 μM EW-7197 before treatment with 100 ng/mL HMGB1. Black arrows: cilia. Scale bar: 2 μm.

**Figure 6 ijms-21-08434-f006:**
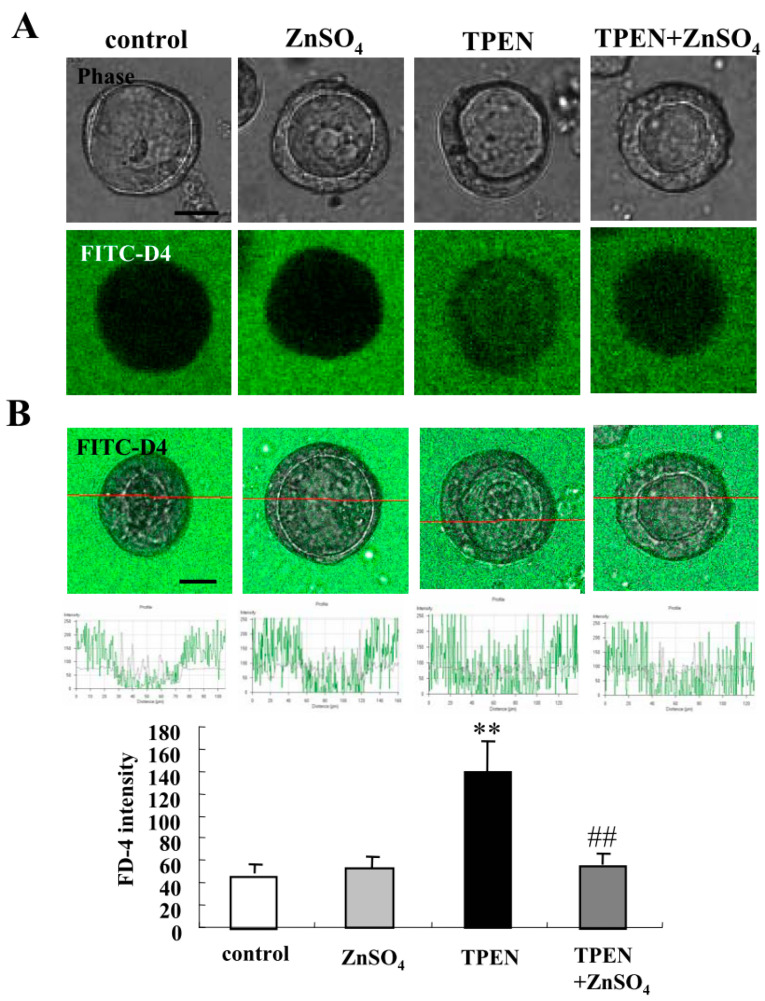
Effects of treatment with ZnSO_4_ and TPEN on epithelial permeability in 2.5D Matrigel culture of Caco-2 cells. (**A**) Phase-contrast images and FD-4 assay of 2.5D Matrigel culture of Caco-2 cells pretreated with 200 nM ZnSO_4_ before treatment with 10 nM TPEN. Scale bar: 20 μm. (**B**) Quantification of FD-4 intensity. Bar graph of FD-4 intensity values representing the barrier function of Caco-2 spheroids pretreated with 200 nM ZnSO_4_ before treatment with 10 nM TPEN. ** *p* < 0.01, vs control, ## *p* < 0.01, vs TPEN. Scale bar: 20 μm.

**Figure 7 ijms-21-08434-f007:**
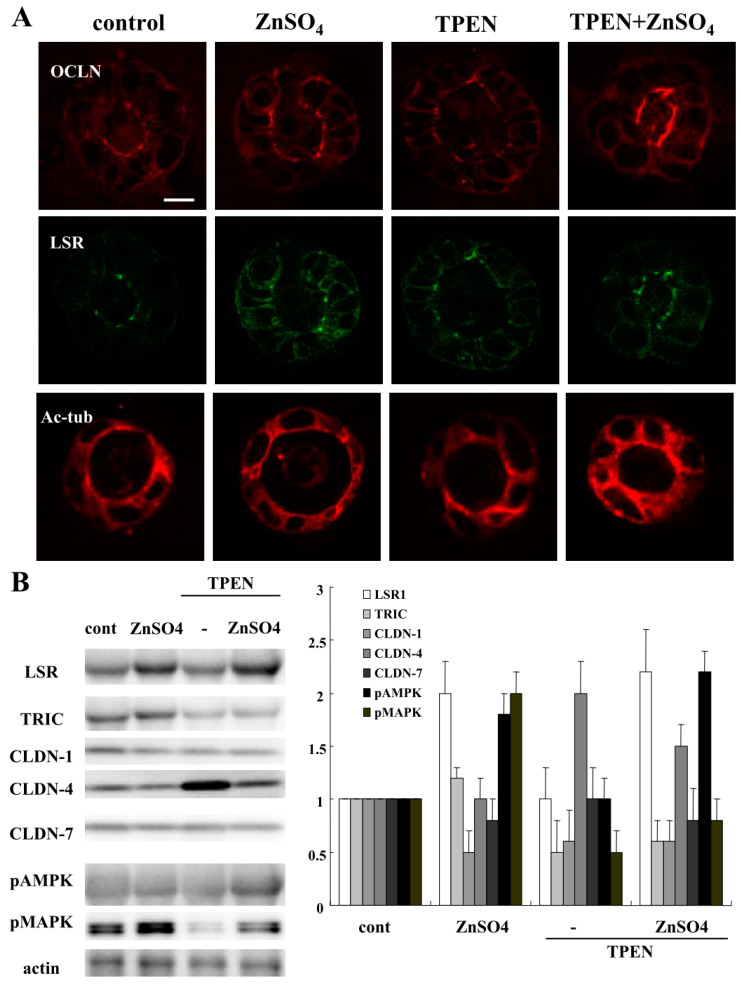
Effects of treatment with ZnSO_4_ and TPEN on tight junction molecules in 2.5D Matrigel culture of Caco-2 cells. (**A**) Immunocytochemistry for OCLN, LSR and Ac-tub in 2.5D Matrigel culture of Caco-2 cells pretreated with 200 nM ZnSO_4_ before treatment with 10 nM TPEN. Scale bar: 20 μm. (**B**) Western blotting for LSR, TRIC, CLDN-1, -4, -7, pAMPK, pMAPK and actin in 2.5D Matrigel culture of Caco-2 cells pretreated with 200 nM ZnSO_4_ before treatment with 10 nM TPEN. The corresponding expression levels of (B) are shown as a bar graph.

**Figure 8 ijms-21-08434-f008:**
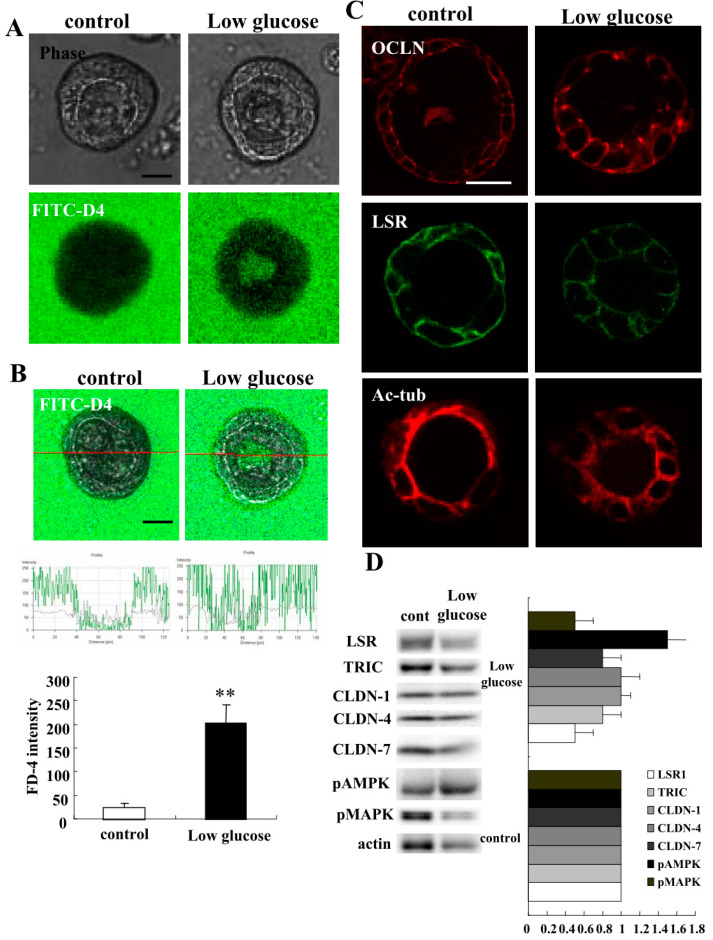
Effects of low glucose medium on epithelial permeability and tight junction molecules treated in 2.5D Matrigel culture of Caco-2 cells. (**A**) Phase-contrast images and FD-4 assay of 2.5D Matrigel culture of Caco-2 cells treated with low glucose DMEM medium. Scale bar: 20 μm. (**B**) Quantification of FD-4 intensity. Bar graph FD-4 intensity values representing barrier function of Caco-2 spheroids treated with low glucose DMEM medium. ** *p* < 0.01, vs control. Scale bar: 20 μm. (**C**) Immunocytochemistry for OCLN, LSR and Ac-tub in 2.5D Matrigel culture of Caco-2 cells treated with low glucose DMEM medium. Scale bar: 20 μm. (**D**) Western blotting for LSR, TRIC, CLDN-1, -4, -7, pAMPK, pMAPK and actin in 2.5D Matrigel culture of Caco-2 cells treated with low glucose DMEM medium. The corresponding expression levels of (D) are shown as a bar graph.

**Figure 9 ijms-21-08434-f009:**
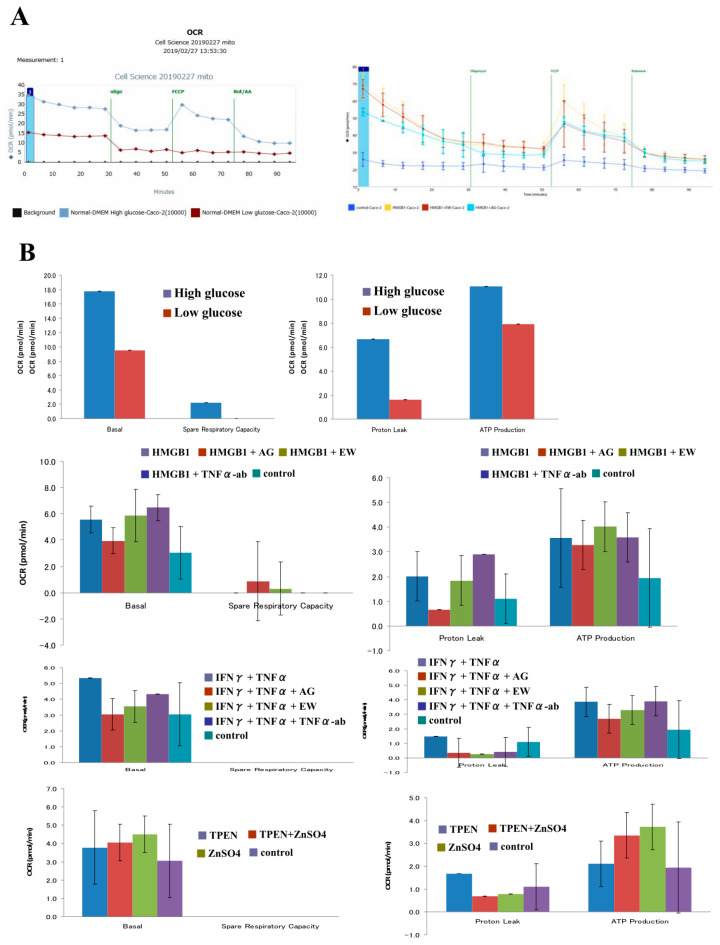
Effects of various treatments on mitochondrial respiration (OCR) in 2D culture of Caco-2 cells. (**A**) Line graph and (**B**) bar graphs of OCR in Caco-2 cells after various treatments. Steady-state OCR measured at six time points. Oligomycin was injected to inhibit ATP synthase, with the addition of FCCP to uncouple mitochondria and obtain the maximal oxygen consumption rate at the 8th time point. Finally, rotenone and antimycin (Rot and AA) were injected to confirm that the respiration changes were due mainly to mitochondrial respiration.

**Figure 10 ijms-21-08434-f010:**
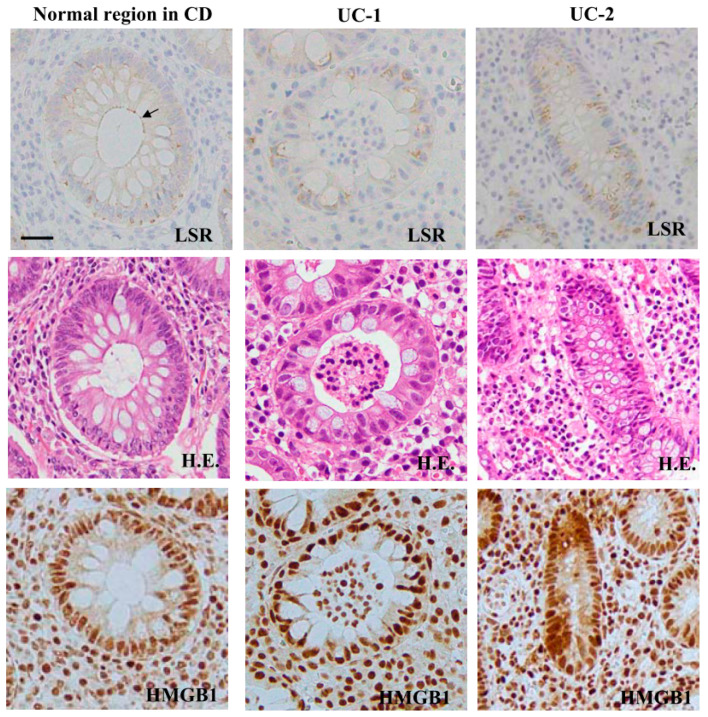
Expression and distribution of LSR and HMGB1 in human normal colonic tissues and inflammatory bowel disease (IBD) tissues. Hematoxylin-eosin (H.E.) staining and immunohistochemical staining for LSR and HMGB1 in normal human colonic tissues and those of the IBD colon. CD: Crohn’s disease. UC: Ulcerative colitis. Bar: 50 μm.

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
