# Peer review of "Increase in Epithelial Permeability and Cell Metabolism by High Mobility Group Box 1, Inflammatory Cytokines and TPEN in Caco-2 Cells as a Novel Model of Inflammatory Bowel Disease"

_ijms, 2020, doi:10.3390/ijms21228434_

Round 1

Reviewer 1 Report

This article is focused on the role of HMGB1 and other pro-inflammatory molecules in the pathogenesis of Inflammatory Bowel disease, and more particularly by increasing  epithelial permeability and mitochondrial respiration of gastrointestinal tract. Moreover, they correlate the zinc deficiency observed in IBD patients with the molecular mechanisms studied in the present work. The article is well structured, the reading is clear and easy to follow and the results obtained are useful and suitable at clinical level. I recommend the article for publication after solve few questions.

Minor comments:

Lines 59-62: Sentence needs to be reviewed.

Line 95: Is HMGB1 concentration used based on previous lab work? Or is it based on a reference? Needs to be clear up.

Figure 2B: Differences between fig. 2B vs 2A and 2C are not clear in the pictures.

Line 130: Same question about concentrations of TNFalpha and IFNgamma and the necessity to include a reference.

Line 182: Include a reference for ZnSO4 and TPEN.

Line 187: The sentence says: “…treatment with ZnSO4 prevented the increase of values induced by TNFalpha and IFNgamma (Figure 6B).”. Nevertheless, figure 6B refers to the effect of ZnSO4 on cells previously treated with TPEN.

Figure 9B: Absence of information about the Control graphed in this figure.

Author Response

Response to Reviewer 1

This article is focused on the role of HMGB1 and other pro-inflammatory molecules in the pathogenesis of Inflammatory Bowel disease, and more particularly by increasing  epithelial permeability and mitochondrial respiration of gastrointestinal tract. Moreover, they correlate the zinc deficiency observed in IBD patients with the molecular mechanisms studied in the present work. The article is well structured, the reading is clear and easy to follow and the results obtained are useful and suitable at clinical level. I recommend the article for publication after solve few questions.

Thank you for your interesting on our experiments. We rewrite it following your suggestions.

Minor comments:

Lines 59-62: Sentence needs to be reviewed.

We added reference [4]. Thus, HMGB1 has the potential to be a therapeutic target for IBD [4]

Line 95: Is HMGB1 concentration used based on previous lab work? Or is it based on a reference? Needs to be clear up.

We previously treated with 100 ng/mL HMGB1 in human lung epithelial cells [12]. In the present study, we also used 100 ng/mL HMGB1 through the experiments. We rewrote them.

Figure 2B: Differences between fig. 2B vs 2A and 2C are not clear in the pictures.

Figure 2B shows opening tight junction structures by HMGB1 in TEM images. Figures 2A and 2C show the changes in expression and localization of tight junction molecules in disruption of tight junctions by HMGB1. All Figures 2A, 2B and 2C indicate disruption of tight junctions by HMGB1.

Line 130: Same question about concentrations of TNFalpha and IFNgamma and the necessity to include a reference.

In our previous report (reference [11]), we treated the cells with 100 μg/mL TNFα and 100 μg/mL IFNγ. In the present study, Caco-2 cells were also treated with 100 μg/mL TNFα and 100 μg/mL IFNγ.

Line 182: Include a reference for ZnSO4 and TPEN.

We added reference [26].

Line 187: The sentence says: “…treatment with ZnSO4 prevented the increase of values induced by TNFalpha and IFNgamma (Figure 6B).”. Nevertheless, figure 6B refers to the effect of ZnSO4 on cells previously treated with TPEN.

We miswrite it. We changes to ''whereas treatment with ZnSO4 prevented the increase of values induced by TPEN (Figure 6B)''.

Figure 9B: Absence of information about the Control graphed in this figure.

We agreed that in some graphs, the value of control was absence. In this experiment by using Extracellular Flux Analyser, the values of control Caco-2 cells were not detected or low level, while the values of all treatment were highly detected. The reasons were unclear.

Reviewer 2 Report

In this study, the authors are focused at understanding the detailed mechanisms underlying the roles of High mobility group box 1 (HMGB1) and zinc deficiency in the intestinal epithelial barrier and cellular metabolism in intestinal inflammation. For the study, the authors utilized Caco-2 monolayer cells (2D culture) and Caco-2 spheroid cells (2.5D Matrigel culture). They showed for the first time that HMGB1 and inflammatory cytokines induced epithelial permeability and cell metabolism concomitant with a downregulation of the tight junction proteins, LSR, TRIC and CLDN-1 in 2.5D culture of Caco-2 cells. TGFbR1 or EGFR kinase inhibitors, (EW-7197 or AG-1478) and TNFα antibody prevented hyperpermeability induced by HMGB1 and inflammatory cytokines (TNFα and IFNγ) in 2.5D Caco2 culture. Similar effects were also observed on cell metabolism altered by HMGB1 and inflammatory cytokines in 2D Caco2 culture. Moreover, ZnSO4 prevented the hyperpermeability induced by the zinc chelator, TPEN in 2.5D Caco2 culture. The manuscript is straight forward and the results are clearly presented and support the conclusions stated. The findings are clinically relevant and suggest that TNFα antibody, EW-7197, AG-1478 (TGFb/EGF signaling) and ZnSO4 may have potential for use as therapeutics against intestinal inflammation such as IBD. However, the authors should consider the following suggestions:

  1. The abstract does not mention studies that show the effects of low glucose conditions on epithelial barrier function and mitochondrial respiration in Caco2 cells.
  2. The concentrations of TNFα and IFNγ seem to be very high? Studies have shown that usually ng/ml concentrations of cytokines are used to examine their effects in intestinal epithelial cells. Why such high supra-physiological conc. of 100mg/ml was used for these studies?
  3. 2 and 4- It is intriguing as to why the house keeping gene actin is also altered (decrease) in response to HMGB1 or TNFα and IFNγ? This could be due to actin being a part of the TJ assembly. The authors could use another house keeping gene, GAPDH. Also, why HMGB1 or TNFα and IFNγ + TNF antibody lane is missing in the western blots in Figs. 2 and 4?  
  4. 7B- Can the authors comment on the upregulation of claudin 4 by the Zn chelator, TPEN as claudin 4 has been shown to be down-regulated in UC.
  5. 8D- The densitometric analysis of the western blot shown is not clear. The bar graphs should be represented in such a way that compares the expression of the target gene in the presence or absence of low glucose.
  6. 9- Not clear if the authors used 2D monolayer Caco2 cells or 2.5D spheroids to examine the effect of TNFa and IFNg on mitochondrial respiration levels.
  7. Does ZnSO4 supplementation also attenuate the inhibitory effects of HMGB1 or TNFα and IFNγ on epithelial permeability and tight junction (TJ) or tricellular TJs (tTJ) proteins?

Author Response

In this study, the authors are focused at understanding the detailed mechanisms underlying the roles of High mobility group box 1 (HMGB1) and zinc deficiency in the intestinal epithelial barrier and cellular metabolism in intestinal inflammation. For the study, the authors utilized Caco-2 monolayer cells (2D culture) and Caco-2 spheroid cells (2.5D Matrigel culture). They showed for the first time that HMGB1 and inflammatory cytokines induced epithelial permeability and cell metabolism concomitant with a downregulation of the tight junction proteins, LSR, TRIC and CLDN-1 in 2.5D culture of Caco-2 cells. TGFbR1 or EGFR kinase inhibitors, (EW-7197 or AG-1478) and TNFα antibody prevented hyperpermeability induced by HMGB1 and inflammatory cytokines (TNFα and IFNγ) in 2.5D Caco2 culture. Similar effects were also observed on cell metabolism altered by HMGB1 and inflammatory cytokines in 2D Caco2 culture. Moreover, ZnSO4 prevented the hyperpermeability induced by the zinc chelator, TPEN in 2.5D Caco2 culture. The manuscript is straight forward and the results are clearly presented and support the conclusions stated. The findings are clinically relevant and suggest that TNFα antibody, EW-7197, AG-1478 (TGFb/EGF signaling) and ZnSO4 may have potential for use as therapeutics against intestinal inflammation such as IBD. However, the authors should consider the following suggestions:

Thank you for your interesting on our experiments. We consider and discuss following your suggestions.

1. The abstract does not mention studies that show the effects of low glucose conditions on epithelial barrier function and mitochondrial respiration in Caco2 cells.

It is known that low glucose condition induces disruption of epithelial barrier and affects mitochondrial respiration by low level of extracellular ATP. In the present study, we use low glucose conditions as positive controls [39]. Accordingly, the effects of low glucose was deleted.

2. The concentrations of TNFα and IFNγ seem to be very high? Studies have shown that usually ng/ml concentrations of cytokines are used to examine their effects in intestinal epithelial cells. Why such high supra-physiological conc. of 100 mg/ml was used for these studies?

In our previous reports, we treated the cells with 100 ng/mL HMGB1, 100 mg/mL TNFa and 100 mg/mL IFNg [11, 12]. In the present study, Caco-2 cells were also treated with 100 ng/mL HMGB1, 100 mg/mL TNFa and 100 mg/mL IFNg.

3. 2 and 4- It is intriguing as to why the house keeping gene actin is also altered (decrease) in response to HMGB1 or TNFα and IFNγ? This could be due to actin being a part of the TJ assembly. The authors could use another house keeping gene, GAPDH. Also, why HMGB1 or TNFα and IFNγ + TNF antibody lane is missing in the western blots in Figs. 2 and 4?  

We agreed the changes of the house keeping gene actin in Figures 2 and 4. We also think that GAPDH is useful for house keeping gene. Figures 2 and 4 are Western blotting images of 2.5D culture of Caco-2 cells. In 2.5D culture, as the numbers of spheroids are different, the values of actin are different. Accordingly, the same values of actin and GAPDH are difficult in all groups. We performed Western blotting of 2.5D culture more than three times.

In the preset study, we focused EW-7197 and AG-1478 for inflammatory conditions. Although we performed Western blotting images of 2.5D culture of Caco-2 cells treated with HMGB1 + TNFα antibody or TNFα and IFNγ + TNFα antibody, in the present study, the changes of bands were not observed (data not shown).

4. 7B- Can the authors comment on the upregulation of claudin 4 by the Zn chelator, TPEN as claudin 4 has been shown to be down-regulated in UC.

We know that claudin-4 and -7 staining was down-regulated in active UC, whereas claudin-2 staining was up-regulated (Oshima T et al., Changes in the expression of claudins in active ulcerative colitis. J Gastroenterol Hepatol. 2008 Dec;23 Suppl 2:S146-50). Zinc deficiency leads to the downregulation of OCLN and ZO-1, and zinc supplementation increases TEER and induces expression of CLDN-2, -7 and ZO-1 (reference [24-26]). Apical Zn rescues claudin-4 relocalization (Sarkar et al., Zinc ameliorates intestinal barrier dysfunctions in shigellosis by reinstating claudin-2 and -4 on the membranes. Am J Physiol Gastrointest Liver Physiol 2019 316: G229 –G246). However, the effects of Zn chelator TPEN on claudin-4 remain yet unclear. We think that TPEN directly may affect claudin-4 expression.

5. 8D- The densitometric analysis of the western blot shown is not clear. The bar graphs should be represented in such a way that compares the expression of the target gene in the presence or absence of low glucose.

We mistook it. We changed the label.

6. 9- Not clear if the authors used 2D monolayer Caco2 cells or 2.5D spheroids to examine the effect of TNFa and IFNg on mitochondrial respiration levels.

We used 2D culture of Caco-2 cells. We added the words.

7. Does ZnSO4 supplementation also attenuate the inhibitory effects of HMGB1 or TNFα and IFNγ on epithelial permeability and tight junction (TJ) or tricellular TJs (tTJ) proteins?

We agreed your suggestions and performed the preliminary experiment. 2.5D culture of Caco-2 cells were pretreated with ZnSO4 before treatment with HMGB1.

Treatment with ZnSO4 prevented the hyperpermeability of FD-4 induced by HMGB1 in the lumina of 2 of 10 spheroids. In Western blotting, treatment with ZnSO4 prevented the changes of LSR, TRIC and CLDN-1 caused by HMGB1 at one time/three times. It is possible that ZnSO4 may prevent the effects of HMGB1, although more studies are necessary and the mechanisms are unclear. We will perform them in near future.